# Strategies to Improve Coverage of Typhoid Conjugate Vaccine (TCV) Immunization Campaign in Karachi, Pakistan

**DOI:** 10.3390/vaccines8040697

**Published:** 2020-11-19

**Authors:** Farah Naz Qamar, Rabab Batool, Sonia Qureshi, Miqdad Ali, Tahira Sadaf, Junaid Mehmood, Khalid Iqbal, Akram Sultan, Noah Duff, Mohammad Tahir Yousafzai

**Affiliations:** 1Department of Pediatrics and Child Health, Aga Khan University Hospital, National Stadium Rd, Aga Khan University Hospital, Karachi City, Sindh 74800, Pakistan; rabab.batool@aku.edu (R.B.); sonia.qureshi@aku.edu (S.Q.); miqdad.ali@aku.edu (M.A.); syeda.sadaf@aku.edu (T.S.); Junaid.mehmood@aku.edu (J.M.); tahir.yousafzai@aku.edu (M.T.Y.); 2Kharadar General Hospital, Agha Khan Road, Nawab Mahabat Khanji Rd, Kharadar Karachi, Sindh 74000, Pakistan; drkhalidiqbal68@gmail.com; 3E.P.I Sindh, Ex I.I Depot Rafiqui Shaheedi Road, Karachi Cantonment, Karachi, Sindh 75510, Pakistan; akramsultandr21@gmail.com; 4Sabin Vaccine Institute, 2175 K Street, NW, Suite 400, Washington, DC 20037, USA; Noah.Duff@Sabin.org

**Keywords:** extensively drug-resistant typhoid, outbreak, mass immunization campaign, typhoid conjugate vaccine

## Abstract

The emergence and spread of extensively drug-resistant (XDR) typhoid in Karachi, Pakistan led to an outbreak response in Lyari Town, Karachi utilizing a mass immunization campaign with typhoid conjugate vaccine (TCV), Typbar TCV®. The mass immunization campaign, targeted Lyari Town, Karachi, one of the worst affected towns during the XDR typhoid outbreak. Here we describe the strategies used to improve acceptance and coverage of Typbar TCV in Lyari Town, Karachi. The mass immunization campaign with Typbar TCV was started as a school- and hospital-based vaccination campaign targeting children between the age of 6 months to 15 years old. A dose of 0.5 mL Typbar TCV was administered intramuscularly. A mobile vaccination campaign was added to cope with high absenteeism and non-response from parents in schools and to cover children out of school. Different strategies were found to be effective in increasing the vaccination coverage and in tackling vaccine hesitancy. Community engagement was the most successful strategy to overcome refusals and helped to gain trust in the newly introduced vaccine. Community announcements and playing typhoid jingles helped to increase awareness regarding the ongoing typhoid outbreak. Mop-up activity in schools was helpful in increasing coverage. Networking with locally active groups, clubs and community workers were found to be the key factors in decreasing refusals.

## 1. Introduction

Typhoid fever, an infection caused by *Salmonella enterica* subspecies *enterica* serovar Typhi (*S*. Typhi) [1,2], is a major cause of febrile illness in many low- and middle-income countries, accounting for an estimated 10.9 million annual infections and more than 116,000 deaths globally [3,4]. The only effective way to treat typhoid fever is with antibiotics, however, the rate of drug-resistant typhoid cases has been increasing globally since the 1990s, particularly in endemic communities in South Asia and sub-Saharan Africa [5].

In Pakistan, the increasing rate of resistance to first- and second-line antibiotics has led to a growing dependence on third generation cephalosporins (ceftriaxone and cefixime) for the treatment of typhoid fever infections [6]. In November 2016, extensively drug-resistant (XDR) S. Typhi—resistant to first- and second-line antibiotics as well as third generation cephalosporins—was reported in Hyderabad, Sindh Province, Pakistan [7]. Over the ensuing months, this new strain quickly spread to the neighboring city of Karachi, where, from January 2017 to June 2019, 6107 cases of blood culture confirmed typhoid was reported. Of these, 63% were XDR [8].

The emergence of XDR typhoid fever in Pakistan has seriously limited the treatment options for the illness, leaving only azithromycin as an effective oral antibiotic treatment option. To reduce the need for antibiotic therapy and reduce the risk of growing drug resistance, prevention of typhoid through vaccination is the most feasible option. [9]. 

In 2017, the World Health Organization pre-qualified the typhoid conjugate vaccine, Typbar TCV® (Bharat Biotech International Limited, Hyderabad, India), recommending that introduction of TCV should be prioritized in settings with a high burden of typhoid illness or drug-resistant strains [10]. Thereafter, the vaccine was first used as part of the response to the rapidly spreading XDR typhoid outbreak in Hyderabad, Pakistan in February 2018, in which Aga Khan University Hospital (AKUH) successfully administered 207,000 doses of Typbar TCV during a mass immunization campaign [11]. Building upon this success, in April 2019 AKUH began a mass immunization campaign in Lyari Town, Karachi, where large numbers of XDR typhoid cases were reported.

While Pakistan’s immunization coverage remains chronically low across all regions, immunization coverage is lowest in Sindh Province, with only 35% of the population receiving immunization as part of Pakistan’s Expanded Programme for Immunization (EPI) [12]. Polio eradication continues to be a national emergency for Pakistan. However, due to suboptimal coverage of the polio vaccine, with significant pockets of continuously missed children despite repeated supplementary polio immunization campaigns, the virus continues to circulate and spread, resulting in recurrent outbreaks [12]. In addition, a lack of community awareness, misconceptions, religious beliefs, illiteracy, rumors, and fears related to vaccines circulating through social media are contributing factors to vaccine hesitancy and resistance within communities [13,14,15].

Typhoid fever most commonly affects children, with 90% of XDR typhoid fever cases in Karachi occurring among children <15 years of age [11]. We started a vaccination campaign aimed to vaccinate 80,000 children between 6 months and 15 years of age with Typbar TCV (intramuscularly as a single 0.5 mL dose) in Lyari Town, Karachi. Here we report the strategies used during the Typbar TCV campaign to improve vaccination coverage in Lyari Town, Karachi, Pakistan.

## 2. Materials and Methods

### 2.1. Setting

Lyari Town, population 846,434, is one of the oldest and most densely populated urban slum settlements in Karachi. Lyari is divided into 11 smaller administrative units, called union councils, which make up the town [16].

The town has a rich political and ethnic culture, yet it has also faced decades of social and political chaos and has been riddled with crime, violence, poverty, overpopulation, and radical politics [17]. This, in turn, has impeded the economic development of basic services. Lyari Town frequently faces acute water shortages, a near total lack of adequate sanitation amenities, prolonged power failures, intolerable housing conditions, illegal encroachment of municipal land, and poorly maintained roadways. Additionally, a concentration of numerous warehouses and industrial units storing and manufacturing hazardous materials have intensified the agonies of the people in this overcrowded locality [16,18]. These conditions contribute to environmentally transmitted infections, such as typhoid fever, in the urban slum population [19,20].

### 2.2. Planning of the Immunization Campaign

The mass immunization campaign was conducted from 10 April 2019 until 24 October 2019. We designed the immunization campaign as a school-based and a hospital-based campaign. The school-based vaccination was conducted at all public and private schools in Lyari Town. The hospital-based vaccination campaign was based at 3 key hospitals serving the population of Lyari Town: Lyari General Hospital, Kharadar General Hospital and Aga Khan Secondary Care Hospital. A mobile vaccination camp was added to target non-school-going young children.

The aim of the school-based campaign was to cover school-going children between 2 years and 15 years of age. The hospital-based campaign was set up at existing EPI centers and hospitals and aimed to cover children of the younger age group who visited the hospitals for their routine childhood immunization and to cover non-school-going children.

#### 2.2.1. School-Based Campaign

The first step for successful school coverage was the identification of all public and private schools in Lyari. Community health workers (CHWs) were trained to record GIS (geographic information system) coordinates of all the schools in Lyari Town and we created a map to help the school-based team navigate all the addresses.

We mapped a total of 436 schools in Lyari Town. The campaign was conducted in 2 phases: pre-vacation and post-vacation. The pre-vacation phase was from 10–30 April 2019, prior to the start of the summer vacation on 4 May 2019, when the school-based vaccination was stopped. During the first phase, a total of 53 schools were covered. The schools reopened on 1 July 2019, and the post-vacation phase of the school-based campaign was conducted from 3 July to 24 October 2019 and covered the remaining schools.

Dates for conducting the campaign at each school were pre-decided after a meeting with the school administration and parental consents were taken prior to the campaign. Consent forms were sent with children for parental signatures and were collected prior to the campaign. On the day of vaccination, teams set up a small vaccination counter where the vaccines, sharp box, cooler, and waste bins were fixed. This counter had tables and chairs for the staff to sit, fill vaccination cards, record the vaccination date and time, and inject the vaccines depending on the strength of the students in the school. Class teachers collected consent forms and made a queue of the children and sent them one by one with their consent forms in hand. School absenteeism in Lyari Town is high and there was limited opportunity for direct interaction with parents. To overcome this challenge, we incorporated a mop-up activity. From 2 April to 24 October 2019, mop-up activities were conducted, which were helpful to some extent in improving the overall vaccination coverage in schools. A list of students who were absent on the day of the vaccination was maintained and a mop-up team revisited the schools and vaccinated those children.

#### 2.2.2. Hospital-Based Campaign

The hospital-based vaccination began in May 2019. We first started at Aga Khan Secondary Care Hospital on 5 May 2019, followed by Kharadar General Hospital and Lyari General Hospital on 15 May and 16 May 2019, respectively. At each health facility, we provided an orientation on the campaign activities to all physicians. During the summer vacation, the school-based vaccination team pivoted activities to actively engage in advocacy and community mobilization, which helped to increase the number of children vaccinated in hospitals in May and June. We visited the local general physicians’ offices and private clinics to share campaign information and to help disseminate the safety awareness messages.

#### 2.2.3. Mobile Vaccination Campaign

The mobile camps helped to engage non-responsive parents and to cover non-school-going children, street children, and child laborers. The mobile vaccination camps began activities on 17 July 2019 and continued until 5 October 2019. Two team members were designated to arrange camps and communicate to the school-based vaccination teams on where to resume community mobilization activities after completing their school targets. Teams went out into the community every day after school to set up camp at places provided voluntarily by community members, such as the houses of community and religious leaders as well as the offices of members of political parties.

The multi-component, three-pronged approach was implemented to reach a larger population that would have otherwise been missed.

## 3. Data Collection Methods

All the data were collected electronically using a web-based application developed by the Data Management Unit at AKU. We had 5 teams of 4 members, including 2 vaccinators, 1 data collector, and 1 team leader who facilitated the activities and school engagement with the administrators and principals. For each child with a consent form, a vaccination card and a vaccinated form (on tablets) was completed. The vaccination form had information on the school, class, age, gender, contact number, address, and GIS coordinates of each child vaccinated.

For each school, data were collected on total enrollments in each class, the number of children vaccinated, absenteeism, and the number of refusals and non-responses. Lists of the absent students from each class were maintained, so that a mop-up team could go back and vaccinate the children who were absent on the day of vaccination. During the school-based vaccination campaigns, the vaccinators were provided with a temperature log sheet to record the temperature every hour to ensure the cold chain storage of the vaccine. Vaccination cards were provided to all vaccinated children. The vaccination form was filled in electronically at the schools, healthcare facilities, and mobile community camps. The GIS coordinates were collected for all vaccinated children to see the location where each child was vaccinated. A live dashboard was available to the research supervisors at AKUH to monitor the daily field activities and vaccination coverage summaries. Periodically, vaccine coverage was calculated at each union council, to help identify the pockets of low coverage. The vaccination plans were guided by the live GIS maps to keep the coverage homogenous across the union councils.

## 4. Ethical Approvals

Ethical approval for the campaign was received from both the Institutional Review Board of AKUH and the National Bioethics Committee. Permission was taken from the Health and Population Welfare Department, Sindh and additional approval was taken from the Director of EPI, Sindh to conduct the vaccination campaign in Lyari. Permissions were sought from the Department of Health and Population Welfare to conduct the campaign at the public sector hospital, Lyari General Hospital. Memorandums of understanding (MOUs) were signed between AKUH and each of the collaborating health facilities. For the school vaccination campaign, permission was sought from the Secretary of the Sindh Education and Literacy Department. Furthermore, permission from the District Municipal Corporation (DMC) was taken to conduct the vaccination campaign at DMC schools in the town.

## 5. Description of Strategies Found to be Effective in Improving Vaccination Coverage

### 5.1. Stakeholder Engagement

The first step of project planning involved face-to-face meetings with the Provincial Minister of Sindh for Health and Population Welfare before the initiation of the immunization campaign in Lyari Town. The Provincial Minister of Sindh for Health and Population Welfare also facilitated and supported the approvals from Drug Regulatory Authority of Pakistan and import of the vaccine from Bharat Biotech Ltd. Several meetings were held with the EPI team, District Health Officer, and Town Health Officer of Lyari Town, and project orientation was provided. Further, this stakeholder group helped us in developing the micro-plan and a social map of Lyari Town as well as adverse events after immunization (AEFI) referrals to the local health facilities.

The Provincial Ministry of Education through the Directorate of the All Private School Management Association Sindh, which supports and monitors the private schools in Pakistan, was also engaged. Furthermore, permissions from the DMC, including both the Medical Officer of DMC South Karachi as well as the Lyari Town Health Officer, were taken and a detailed plan of the activity was shared with them. This agreement and cooperation were mandatory to gain access to the community and to enable successful project implementation of the vaccination campaign at DMC schools in town. Later, the individual administrators of public and private schools were contacted, and their consent was obtained for conducting the vaccination campaign at each of their respective schools.

### 5.2. Vaccine Education Sessions in Schools and at Religious Institutions

Permissions from all stakeholders including school administrators were taken prior to the planning of the campaign. However, acquiring parental consent was challenging. The school administrators were not engaged in speaking to the parents about the significance of the vaccine and parents were unable to understand the context of the situation, children either forgot to give the consent forms to their parents or some older children even threw away the consent forms to avoid getting a “needle”. We strategized by engaging school principals and teachers and led health awareness sessions for school administrators, principals, and teachers, educating them about the ongoing typhoid fever outbreak and the importance of vaccination. We also delivered health education lectures to the students regarding prevention of typhoid fever and health hygiene practices, such as hand washing.

At the schools where administrators cooperated, interactive sessions with groups of parents were also held and their concerns regarding the vaccine were addressed to gain their confidence.

Some of the children in Pakistan go to religious institutions, called “madrassas so to cover these institutions meetings were held with religious stakeholders, such as khateebs and imams (religious leaders), in mosques to seek permission to vaccinate the children within these institutions. We identified 50 madrassas in Lyari Town, however we could cover only 4 of them due to time constraints, as the national TCV campaign was already planned. The list of remaining madrassas was shared with the EPI for the upcoming national campaign.

### 5.3. Community Engagement and Social Media Campaign

The community-based vaccination involved house-to-house canvassing and distributing handbills and pamphlets on typhoid fever and the placement of banners and pamphlets at prominent community places. Street and market announcements were made using megaphones. Pamphlets and banners were displayed at popular community locations, such as outside of youth centers, football clubs, restaurants, cafes, and markets. Social activist and community groups were identified and approached, and political and community leaders were asked to arrange community meetings at their offices to influence people to get their children vaccinated. Influential religious stakeholders were identified, and meetings were held with religious leaders, including khateebs and imams. Where allowed, announcements in mosques were made. Megaphones playing a typhoid jingle along with campaign announcements were placed in vehicles, which were driven throughout the areas adjacent to the community camps. The mobile vaccination camps provided an opportunity for underserved populations to get their children vaccinated and allow CHWs to directly interact with parents to address their questions and concerns.

Various WhatsApp and Facebook groups were identified, and their administrators were approached to disseminate awareness messages on their respective platforms. Aga Khan Secondary Care Hospital sent mobile text messages to a population of 50,000 living in Lyari Town. Different sports, lingual, and religious groups, as well as local political parties and social work organizations working in Lyari Town, supported and helped to expand the visibility and community acceptance of the campaign activities. All these platforms enabled the successful implementation of the vaccination campaign.

### 5.4. Staff Trainings and Field Supervision

Trainings were held on the importance of the TCV catch-up campaign and its key strategies to achieve the target, roles, and responsibilities of different cadres engaged and involved during the vaccination campaign. Trainings were also held on reporting and recording tools, use of supervision and monitoring tools, vaccine administration and infection prevention, waste management protocols and interpersonal communication, and social mobilization to guide parents and caregivers.

Medical staff members of the team (2 doctors and 2 nurses) were trained on adverse events following immunization (AEFI) management and essential referral and response protocols, and each team was provided with AEFI kits for AEFI management. The teams provided a 24-h hotline number for the reporting of AEFI to each vaccinated child on the back of the vaccination card. We worked in continuous collaboration with EPI and district health officers assisted us in AEFI monitoring and reporting and identified hospitals for referral in case of any adverse events. This helped us gain the trust of communities and helped us at all levels where safety concerns regarding vaccine were raised. A single adverse event may create rumors in the community that may endanger the whole campaign [21]. A crisis management plan with communication strategies was prepared and shared with team members to prevent rumors from jeopardizing the campaign [21]. Each team leader was trained to assure the safety and security of field staff during field activities.

### 5.5. Healthcare Provider Education 

Since this was a new vaccine and physicians were not aware of it, we mapped and visited local general physicians’ offices and private clinics to help disseminate the safety awareness messages.

We visited all local healthcare clinics in Lyari Town and educated the healthcare providers about the ongoing typhoid fever outbreak and immunization campaign. Information, education, and communication (IEC) materials were developed in the local language, Urdu, and posted in the waiting areas of the local general physicians’ offices to increase patients’ awareness. Pamphlet stands with information to bring children for vaccination were placed outside the health facilities.

The queries of healthcare providers regarding the vaccine were addressed on a regular basis. This proactive approach to communication helped us to avoid any misinformation from spreading. Vaccination was conducted at those facilities, where allowed.

## 6. Results

A total of 324 schools and 39,939 children were covered during the school-based vaccination campaign. The acceptance rate in school was 48.2% (total children vaccinated in schools: 44,993; total enrollments in schools: 93,176). Mop-up activities were conducted in 99 schools, which increased school-based coverage by 17% (from 36% to 53%). Through the mop-up activities we vaccinated an additional 5054 children. In total, 12,101 (26.9%) children were vaccinated in government schools, 31,578 (70.2%) were vaccinated in private schools, and 1314 (2.9%) were vaccinated in madrassas (Table 1).

In total, 16,042 children were vaccinated in healthcare facilities through the hospital-based campaign and 26,958 children were vaccinated in the mobile vaccination camps. Overall, 30% of children were vaccinated in community camps (Table 2).

Children younger than 3 years of age were targeted through hospital- and community-based campaigns, while older children were targeted through schools (Table 3).

Estimated coverage of Typbar TCV was calculated among children 6 months to 15 years of age in all the union councils of Lyari Town, Karachi (Figure 1).

## 7. Discussion

Several strategies were implemented to address, respond, and manage the issue of low vaccine coverage and to identify strategies for improving vaccination coverage in a mass immunization campaign in Lyari Town, Karachi. The use of these strategies helped to maximize the vaccine coverage.

The use of multi-component interventions, including stakeholder engagement, health provider and parental education, community mobilization, use of social media, and inclusion in a community-based campaign have been proven to be effective strategies in improving the coverage of newly introduced vaccines [22,23,24,25]. We similarly chose a multi-component approach, incorporating various strategies and activities that were appropriate for the local context. Before the implementation of the newly introduced vaccine, the local context of the slum area was carefully considered. A review of new and existing activities and strategies to successfully address vaccine hesitancy was used to help prioritize the project activities and strategies based on an evaluation of their potential impact. The interventions we implemented, including awareness sessions at schools, community meetings, and mobile camps, aimed to directly target non-vaccinated or under-vaccinated populations; increase knowledge and awareness regarding vaccination; improve ease and access to the vaccine; target local community and healthcare workers; and engage with religious or other influential leaders to encourage vaccination in the community. Awareness strategies aimed to address the local population and its specific concerns to reduce the information gaps were also instrumental. The advocacy and communication strategies were evolved based on the day-to-day experiences and lessons learnt as the project moved ahead.

Community feedback was also an important factor, and helped us to understand the barriers to vaccination. During the pre-vacation school-based vaccination campaign, we added a clause to the consent forms for parents who refused vaccination to provide the reason for not vaccinating their child. Some parents mentioned on the consent forms that their children had completed their vaccination as per the EPI schedule. The reasons provided by parents were helpful in counselling the parents and to include responses to these concerns in the awareness sessions. The introduction of education initiatives, such as the education sessions with school administrators, teachers, and parents, was found to be an effective intervention and to increase knowledge and awareness and change attitudes.

During the school vacation period, we deployed our school-based vaccination teams throughout the streets and markets within the community to make announcements and distribute handbills about the hospital- and mobile-based vaccination campaigns. We advised our team to follow up after distributing handbills, allowing the community members time to consider the information, ask questions, and have their queries answered. Additionally, announcements were made in the local mosques following Friday prayers to target a large segment of the population near the community camps. However, not all mosques allowed the announcements. To cope with this challenge, we provided our teams megaphones to be placed in the mobile vans which were driven throughout the nearby residential area of the community camps playing a typhoid jingle in the local language—a strategy found to be successful in increasing vaccine uptake in Malawi [26]. This significantly increased the number of children vaccinated in the community camps.

Considering the high absenteeism from the school and the limited to no interaction with parents in the school-based campaign, community camps and mobile van interventions were key not only to increase the vaccination coverage, but also to serve as a source of community engagement and advocacy and as a means of communication. This intervention provided an opportunity for two2-way dialogue with parents and the general public. The mobile van concept was an unconventional concept similar to the “Vaccination-on-Wheels” clinic [27,28,29] with the aim of “reaching the bottom of the pyramid.” Lack of access is still considered the biggest barrier hindering vaccine coverage [30,31,32]. The mobile van and community camps provided a vaccination service free of cost and close to home, and assured equitable access of the vaccine to the most impoverished. We vaccinated non-school-going children, children in difficult-to-reach areas, street children, and child laborers with the help of the mobile van. When people in the community heard the announcements and the typhoid jingle from the mobile van and community camps, they came out of their homes and approached the mobile team to ask about the activities, providing us with an opportunity to directly interact and counsel them. This intervention as a whole served as a promotional activity, particularly in the context of the introduction of a new vaccine in real field settings. Additionally, given the high rates of school absenteeism in Lyari Town, we initiated mop-up activities where the school-based vaccination team maintained a list of absent students and would later revisit the schools and vaccinate those who were absent on the last day of vaccination.

Healthcare providers play an important role in the community to build confidence for newly introduced vaccines, as they are often the most reliable advisors and primary persuaders of vaccination decisions in the community [33]. Knowledge about particular vaccines, including their efficacy and safety, helps to build physicians’ own confidence in vaccines and their willingness to recommend them to others. Because of this, we mapped all of the local healthcare clinics in Lyari Town and incorporated activities, such as healthcare provider education sessions, to increase their awareness and knowledge about Typbar TCV.

Use of digital media has significant effects on healthcare utilization [34,35]. Use of large-scale mobile messages and mobile apps have been reported to be effective mediums to increase vaccination coverage, including in previous vaccination campaigns in Matiari, Pakistan [36,37]. Distributing reminders via postal mail, call, text messaging, and email were found to be effective in increasing vaccination coverage in Nigeria, Guatemala, and San Diego [38,39,40,41]. Another study reported that text message reminders improved the timely MMR vaccination of high-risk children in New York City [42]. Since 97% of Pakistan’s population owns a mobile phone, text and voice messages are readily available, accessible, and inexpensive means to promote the introduction of new vaccines in the community. We identified and targeted multiple local WhatsApp and Facebook groups to reach as many people as possible. Some of the participating schools with their own Facebook pages uploaded pictures and videos of the vaccination activities at their school, which helped us to gain cooperation from other schools, as well. Altogether, our use of a multi-component approach successfully increased vaccine uptake, knowledge, and awareness, and altered local attitudes towards vaccination.

The comprehensive training of vaccination teams on interpersonal communication, safe injection practices in field and field-based monitoring, and supportive supervision was helpful throughout, especially in term of gaining the confidence of the communities and ensuring high-quality service delivery outcomes. 

By utilizing a multi-component campaign approach and incorporating several strategies to raise awareness and increase engagement at multiple levels of the community, we were able to successfully reach our target immunization coverage rate as well as earn the community’s trust, cooperation, and support for typhoid vaccination. This approach allowed for a genuine 2-way exchange of ideas with community members, providing us with valuable feedback to help us better understand what approaches are the most appropriate for the local context. While many lessons were learned throughout the duration of the campaign (Table 4), the biggest takeaway is that every interaction can serve as an intervention, and community engagement is an essential component of any immunization campaign.

## 8. Conclusions

The mass immunization campaign of Typbar TCV in Lyari Town confirms the significance of community engagement on the successful implementation of vaccination campaigns in an urban slum setting. Community engagement activities serve to increase awareness, build trust, help increase an understanding of the barriers for new vaccines, and set strategies for increasing vaccine uptake and improving coverage. Every interaction with the community at any point serves as an intervention for the successful implementation of strategies to increase vaccination coverage. Community engagement should be a two-way exchange of ideas between health professionals and the community members, which should guide them to make their own decisions regarding vaccination. However, multiple interactions at different layers of the community are observed to be persuasive.

## Figures and Tables

**Figure 1 vaccines-08-00697-f001:**
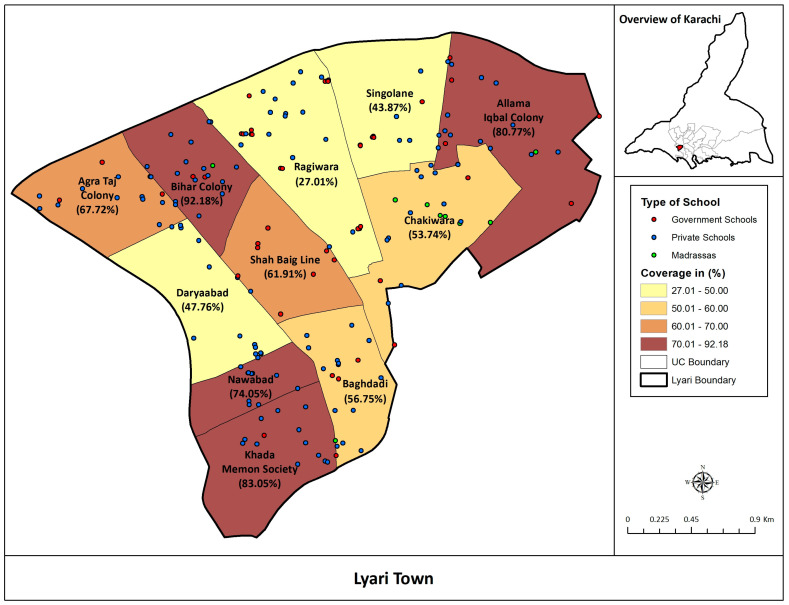
Estimated Coverage of Typbar TCV among children 6 months to 15 years of age in Lyari Town.

**Table 1 vaccines-08-00697-t001:** Number of children vaccinated through the school-based vaccination campaign in Lyari Town.

Union Council	Number of Children Vaccinated in Government Schools	Number of Children Vaccinated in Private Schools	Number of Children Vaccinated in Madrassas	Total; *n* (%)
Agra Taj Colony	1344	5505		6849 (15.2%)
Allama Iqbal Colony	821	2025		2846 (6.3%)
Baghdadi	1351	5036		6387 (14.2%)
Bihar Colony	1695	4147	139	5981 (13.3%)
Chakiwara	1070	2388	861	4319 (9.6%)
Daryaabad	252	1608		1860 (4.1%)
Khada Memon Society	678	4709		5387 (12.0%)
Nawabad	1078	4090		5168 (11.5%)
Ragiwara	1449	1012		2461 (5.5%)
Shah Baig Line	1048	372	84	1504 (3.3%)
Singolane	1315	687	229	2232 (5.5%)
Total number of children vaccinated	12,101 (26.9%)	31,578 (70.2%)	1314 (2.9%)	44,993 (100%)

**Table 2 vaccines-08-00697-t002:** Total number of children vaccinated in Lyari Town using different strategies.

Vaccination Strategy	*n* (%)
Children vaccinated in school-based vaccination campaign	39,939 (45.39%)
Children vaccinated in mop-up activity	5054 (5.74%)
Children vaccinated in hospital-based vaccination campaigns	16,042 (18.23%)
Children vaccinated in community based-vaccination campaigns	26,958 (30.64%)
Total number of children vaccinated	87,993

**Table 3 vaccines-08-00697-t003:** Age breakdown of the children vaccinated in Lyari Town.

Age Groups, Years	School-Based Vaccination Campaign; *n* (%)	Hospital-Based Vaccination Campaign; *n* (%)	Community Based-Vaccination Campaign; *n* (%)	Total; *n* (%)
<3	222 (0.5)	3728 (23.2)	4083 (15.1)	8033 (9.1)
3–6	5873 (13.1)	4212 (26.3)	6186 (22.9)	16,271 (18.5)
6–9	11,356 (25.2)	3520 (21.9)	6284 (23.3)	21,160 (24.0)
9–12	12,784 (28.4)	2634 (16.4)	5789 (21.5)	21,207 (24.1)
12–15	14,758 (32.8)	1948 (12.1)	4616 (17.1)	21,322 (24.2)
Total	44,993	16,042	26,958	87,993

**Table 4 vaccines-08-00697-t004:** List of strategies used and suggested recommendations.

1. Before implementation of any mass immunization program evaluate the strategies best suited for the local context.
2. In the case of urban slum areas, pamphlets, handbills, banners and announcements alone are not enough, opportunities for open, direct communication with parents and children is important.
3. Multiple permanent, temporary, and mobile vaccination posts enable widespread reach throughout the population.
4. Continuous data analysis and periodical calculation of coverage in targeted areas and neighborhoods helps to identify pockets of low-coverage and prioritize further activities in targeted populations in a timely manner.
5. The use of technology may assist not only in sharing messages for vaccination, but for real time data collection for timely action.
6. Bringing all community stakeholders on board may help implement field operations in high-refusal areas.
7. Microplanning should include social mapping of local influencers in the community.
8. A strong AEFI management and referral plan should be in place as even a single SAE in a politically unstable setting can be disastrous for the entire campaign.
9. Effective use of technology is the cheapest way to promote vaccination and enhance awareness.
10. Community members should be asked for their assistance, suggestions, and feedback.
11. Safety and security of field staff in the community should be assured.
12. Discussions and engagement of physicians is mandatory to impart the correct messages to the community.

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
