# Peer review of "Strategies to Improve Coverage of Typhoid Conjugate Vaccine (TCV) Immunization Campaign in Karachi, Pakistan"

_vaccines, 2020, doi:10.3390/vaccines8040697_

Round 1

Reviewer 1 Report

Thank you for the opportunity to review this interesting paper.  I would welcome publication subject to addressing some important issues.  First and most important is the language of the attempted communication; firstly it is a long time since I last visited this area but my recollection is that despite Urdu being the official language, the people in Lyari predominantly speak Balochi as their first language.  At what reading age were the information sheets targetted?  A common problem with health professional generated materials is that the language is too sophisticated for the target population.  Was this addressed or assessed?  

As is mentioned a major finding from other studies is that the way to gain community cooperation with health initiatives is to target community leaders and bring the community behind the initiative.  This is usually far more effective that information leaflets and direction from health professionals. 

With these issues clarified, I would welcome publication. 

Author Response

  1. Thank you for the opportunity to review this interesting paper.  I would welcome publication subject to addressing some important issues.  First and most important is the language of the attempted communication; firstly it is a long time since I last visited this area but my recollection is that despite Urdu being the official language, the people in Lyari predominantly speak Balochi as their first language. At what reading age were the information sheets targetted? A common problem with health professional generated materials is that the language is too sophisticated for the target population.  Was this addressed or assessed?  

Response: True, Balochi is the local language in Layri, however Urdu is widely spoken and understood. The information sheet was targeted for parents and school administrators and teachers. The targeted age was above 15 years and above. The information sheet was translated to Urdu in simple and easy to read language with pictures to make messages clearer. Several meetings were held with the stakeholders and communication material was presented to them and permission was taken from the distribution of Information, Education and Communication (IEC) material in their schools, hospitals and community. The generated material was assessed by Aga Khan Hospital Communication department for appropriateness.

  1. As is mentioned a major finding from other studies is that the way to gain community cooperation with health initiatives is to target community leaders and bring the community behind the initiative.  This is usually far more effective that information leaflets and direction from health professionals. With these issues clarified, I would welcome publication. 

Response: Yes. We observed community engagement and direct communication with different stake holders and community members was more effective than IEC material.

Reviewer 2 Report

Authors present their regional typhoid vaccination and associated community engagement programme. This is a well-written and interesting article; although, I think this report merits a more detailed presentation of the results of this broad and impressive vaccination campaign. I have detailed my suggestions below.

Introduction

  • Line 38 — remove “systematic”
  • Line 38 — genus and species need to be italicised
  • Line 97 — Replace “camp” with “clinic”
  • Line 116 — what is a “coalman”?
  • Line 195 – “obtaining informed parental consent was challenging” rather than a “problem”

Results

  • Further results could be presented here …
  • What was the acceptance rate in the schools if the denominator can be worked out (total children in these schools).
  • What was the age breakdown of vaccinated children?
  • Did you measure any indicators of socioeconomic status that could also be used to stratify the data?
  • Maybe stratify by public/private school if individual socioeconomic measures not available?
  • GIS data is present — is it possible to show the number of vaccinated children by region of schools — line 105 says a map was made can the number of vaccines given be superimposed on this map? This could be a nice figure …
  • Is it possible to present any community feedback — if this has been analysed?

Author Response

  1. Introduction
  • Line 38 — remove “systematic”

Response: The word systematic has been removed from line 38.

  • Line 38 — genus and species need to be italicized

Response: Genus and species have been capitalized in line 38.

  • Line 97 — Replace “camp” with “clinic”

Response: Thank you so much for indicating an important concern. There’s a minor difference between camp and clinic. However, in our vaccination campaign context all authors agree to use the word camp. As we provided more of preventive services rather than clinical i,e. vaccination; we would keep the word camp instead of clinic. Many health programs exhibit one of three common models. Health camps are large-scale initiatives that have more of a preventative rather than curative focus. Like those run by the Afya Kenya Foundation, health camps aim to provide basic health education, preventative care, and health screenings to a wide portion of the population. The second type is the mobile clinic, which is usually smaller and offers basic services like those of a primary health facility.

Link: https://healthmarketinnovations.org/approaches/mobile-clinic

  • Line 116 — what is a “coalman”?

Response: We are sorry for the inconvenience. That’s a mis-spell. The word coleman is changed to coleman in line 116.

  • Line 195 – “obtaining informed parental consent was challenging” rather than a “problem”

Response: The word “a problem has been replaced with “challenging” in line 195.

Results

  1. Further results could be presented here …

Response: Suggested tables and figure have been added.

  1. What was the acceptance rate in the schools if the denominator can be worked out (total children in these schools).

Response: The acceptance rate in school was 48.2 % (total children vaccinated in schools, 44993 / total enrollments in schools, 93176). Text added (line 263 -264).

  1. What was the age breakdown of vaccinated children?

Response: Text and Table 3 – Age breakdown of the children vaccinated in Lyari Town; incorporated in text (line 276 – 278).

  1. Did you measure any indicators of socioeconomic status that could also be used to stratify the data?

Maybe stratify by public/private school if individual socioeconomic measures not available?

Response: We didn’t take socioeconomic data.  Stratification has been done by public private school and text and table 1 has been added to the document (please refer to line 266-270)

  1. GIS data is present — is it possible to show the number of vaccinated children by region of schools — line 105 says a map was made can the number of vaccines given be superimposed on this map? This could be a nice figure …

Response: Thank you so much for your valuable suggestion. Figure 1: Estimated Coverage of Typbar – TCV in all Union Councils of Lyari Town, has been added to the paper showing the coverage in each UC and superimposed by the school GIS map (line 279-283)

  1. Is it possible to present any community feedback — if this has been analysed?

Response: Community feedback wasn’t our primary objective, initially we collected some data but it’s not analyzed yet.